# Mechanism of allosteric inhibition of human p97/VCP ATPase and its disease mutant by triazole inhibitors
Purbasha Nandi [1,2,10,16], Kira DeVore[1,2,16], Feng Wang[3,16], Shan Li[3], Joel D. Walker[4,5], Thanh Tung Truong [4,5,11], Matthew G. LaPorte[4,5], Peter Wipf [4,5], Heidi Schlager[6], John McCleerey [6,12], William Paquette [6], Rod Carlo A. Columbres[3,13], Taiping Gan[3], Yu-Ping Poh[2,14], Petra Fromme[1,2], Andrew J. Flint[7], Mark Wolf[6], Donna M. Huryn[4,8,15], Tsui-Fen Chou [3,9] ✉ & Po-Lin Chiu [1,2] ✉

Human p97 ATPase is crucial in various cellular processes, making it a target for inhibitors to treat cancers, neurological, and infectious diseases. Triazole allosteric p97 inhibitors have been demonstrated to match the efficacy of CB-5083, an ATP-competitive inhibitor, in cellular models. However, the mechanism is not well understood. This study systematically investigates the structures of new triazole inhibitors bound to both wild-type and disease mutant forms of p97 and measures their effects on function. These inhibitors bind at the interface of the D1 and D2 domains of each p97 subunit, shifting surrounding helices and altering the loop structures near the C-terminal α2 G helix to modulate domain-domain communications. A key structural moiety of the inhibitor affects the rotameric conformations of interacting side chains, indirectly modulating the N-terminal domain conformation in p97 R155H mutant. The differential effects of inhibitor binding to wild-type and mutant p97 provide insights into drug design with enhanced specificity, particularly for oncology applications.

Human p97, also known as valosin-containing protein (VCP), is an essential AAA+ (ATPases Associated with various cellular Activities) protein that hydrolyzes ATP to drive diverse cellular functions[1–4] including protein quality control through the ubiquitin-proteasome system (UPS)[5], endosomal sorting[6,7], autophagy[6–8], chromosome condensation[9], and membrane fusion of cellular organelles such as Golgi apparatus, endoplasmic reticulum, and the nuclear envelope[10–13]. p97 orchestrates these functions using a wide array of protein cofactors that engage in a series of protein-protein interactions to carry out these cellular processes[14] and collectively contribute to maintaining normal cellular homeostasis.

Dominantly inherited missense mutations in p97 are associated with a rare multisystem degenerative disorder known as inclusion body myopathy with early-onset Paget disease and frontotemporal dementia (IBMPFD)

which is characterized by four main phenotypes: inclusion body myopathy (muscle), Paget's disease of bone (bone), frontotemporal dementia (brain), and familial amyotrophic lateral sclerosis (fALS)[15,16]. To date, more than 60 missense mutations have been identified in patients with IBMPFD/ALS[17–19]. These mutations are primarily located in the N-terminal domain (NTD) or the interface between NTD and D1 domains of p97[20]. R155H is the most common mutation, being found in 50% of the affected patients[16]. This disease mutant of p97 exhibits a higher ATPase activity than the wild-type enzyme in vitro[21], and the abnormal ATPase activity results in impairments in the UPS pathway[22], endosomal trafficking[23], autophagy[6,8], and mitochondrial function[24,25]. Unfortunately, effective treatments for reducing the elevated ATPase activity of p97 disease mutants or restoring activity to basal levels remain elusive. In addition, the mechanism by which the mutated

[1]School of Molecular Sciences, Arizona State University, Tempe, AZ, USA. [2]Biodesign Center for Applied Structural Discovery, Arizona State University, Tempe, AZ, USA. [3]Division of Biology and Biological Engineering, California Institute of Technology, Pasadena, CA, USA. [4]University of Pittsburgh Chemical Diversity Center, University of Pittsburgh, Pittsburgh, PA, USA. [5]Department of Chemistry, University of Pittsburgh, Pittsburgh, PA, USA. [6]Curia Global, Albany, NY, USA. [7]Leidos Biomedical Research, Inc., Frederick National Laboratory for Cancer Research, Frederick, MD, USA. [8]Department of Pharmaceutical Sciences, University of Pittsburgh School of Pharmacy, Pittsburgh, PA, USA. [9]Proteome Exploration Laboratory, Beckman Institute, California Institute of Technology, Pasadena, CA, USA. [10]Present address: Biology Department, Brookhaven National Laboratory, Upton, NY, USA. [11]Present address: Faculty of Pharmacy, Phenikaa University, Hanoi, Vietnam. [12]Present address: Graduate School of Arts and Sciences, Boston University, Boston, MA, USA. [13]Present address: Center for Cancer Research, National Cancer Institute, National Institute of Health, Bethesda, MD, USA. [14]Present address: Biodesign Center for Mechanism of Evolution, Arizona State University, Tempe, AZ, USA. [15]Present address: Department of Chemistry, University of Pennsylvania, Philadelphia, PA, USA. [16]These authors contributed equally: Purbasha Nandi, Kira DeVore, Feng Wang. ✉e-mail: tfchou@caltech.edu; plchiu@asu.edu

**Fig. 1 | Structure of human p97 ATPase and designed triazole inhibitors used in this study.**
**a** Primary structure of human p97 ATPase. Location of the pathogenic mutation R155H (blue) and arginine fingers of D1 (R359, R362) and D2 domains (R635, R638) are labeled. α/β and α subdomains of the D1 and D2 ATPase domains are underlined. Highlighted regions (pink; D1 α and D2 α/β subdomains) are the domains that interact with the triazole-based inhibitors. **b** Structure of wild-type p97 with ADP and the triazole-based inhibitor bound (PDB code: 8UV2). The N-terminal (NTD), D1, D2, and linker domains are in green, orange, orange-red, and purple, respectively. Bound nucleotides and the triazole-based inhibitor are shown in space-filling representation in light and medium blue, respectively. The helices of the D2 domain are labeled. **c** Chemical structures of the lead compounds for p97 ATPase: NSC799462, NSC804515, NSC819701, NMS-873, and UPCDC30766.

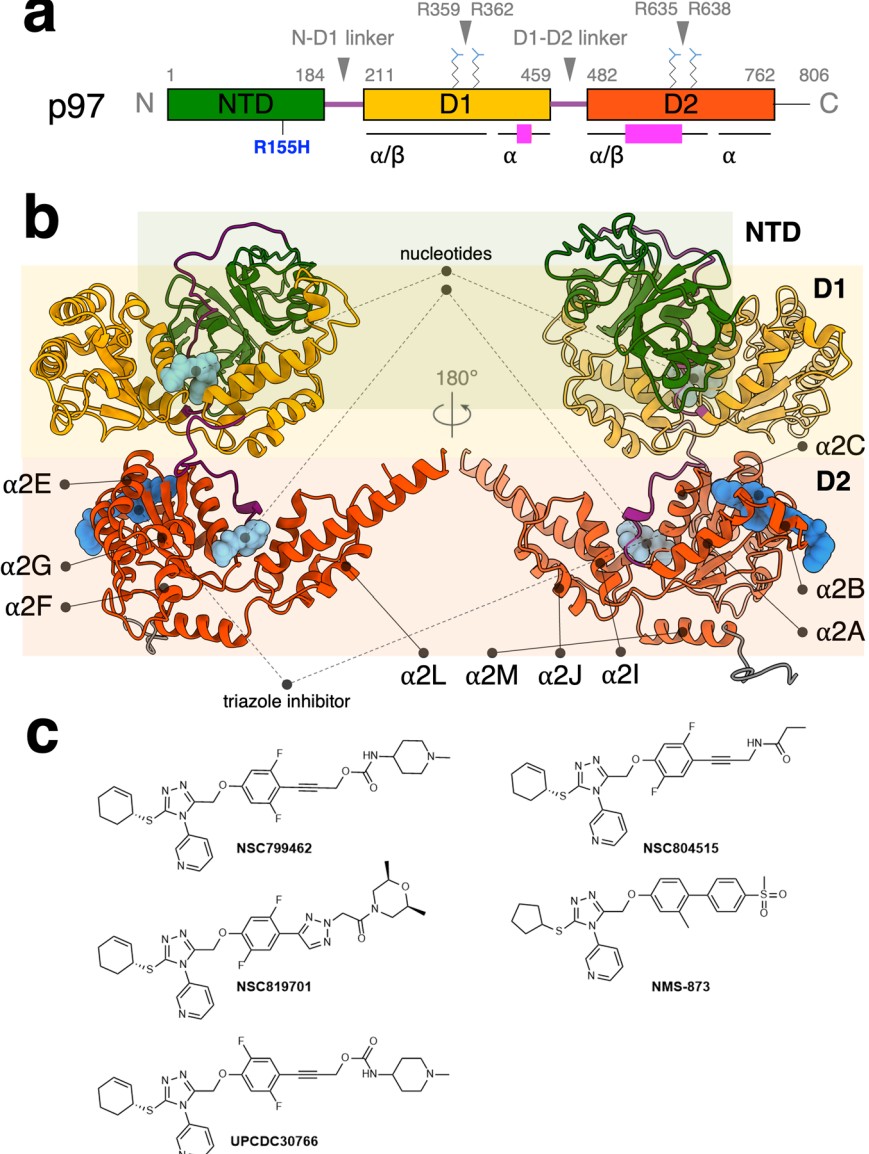

residues lead to abnormal ATPase activity remains unclear. Given that p97 activity is crucial for cell viability, attention should be given to the development of selective modulators of mutant p97 activity for a safe IBMPFD therapy. Also, inhibitors of p97 can induce cellular apoptosis, thus positioning p97 ATPase as a rational drug target for cancer treatment[26–28].

p97 shares a molecular architecture akin to other type II ATPase members in this class of chaperones[29]. The molecular mechanism of p97 ATPase has been extensively studied with a variety of methods, including X-ray crystallography, nuclear magnetic resonance (NMR) spectroscopy, and single-particle cryogenic electron microscopy (cryo-EM)[29]. These approaches have revealed a dynamic molecular machine with three distinct functional domains: an NTD and two ATPase domains, termed D1 and D2[29–31], each harboring conserved Walker A and B motifs typically associated with ATP binding and hydrolysis (Fig. 1a, b)[30–32]. In p97, the D2 ATPase domain is responsible for most of the ATPase activity[33]. A stable and functional oligomer of six p97 monomers assembles without a requirement for binding nucleotides[34]. This homohexamer structure appears as two concentric rings, formed by the D1 and D2 domains, that create a central pore capable of binding protein and peptide substrates for segregation or degradation[35]. Recent structural studies posit an asymmetrical model wherein the substrate traverses the p97 central pore in a hand-over-hand manner[36,37].

The p97 NTDs play a crucial role in binding various cofactors that direct p97's activity toward regulating different cellular functions[14]. The NTDs exist in either of two distinct conformations: the "up" and "down" states[38,39], associated with the occupancy of D1 and D2 nucleotide-binding sites, even though they are far away from each other[40,41]. "Down" NTDs are coplanar with the D1 ring, whereas "up" NTDs protrude above the plane of the D1 ring[30]. The conformational changes of the NTDs are controlled by cofactor binding and the D1 nucleotide-bound states[21,30,42–46]. Notably, the NTDs of the p97[R155H] mutant were found in either up[40,47–52] or down[53] conformation.

Domain-domain communication is crucial for p97 function. The N-D1 and D1-D2 linkers play a pivotal role in regulating the activities of the D1 and D2 ATPase domains (Fig. 1a, b)[29,41]. Nucleotides bind at the interface between the α/β and α helical subdomains and contact the arginine fingers on the second region of homology (SRH) motif of an adjacent subunit (Fig. 1a, b)[3,36]. Mutations in the NTD or binding of protein cofactors modulate the ATPase activity through sensor domains and arginine fingers[14,21,47]. The ability of p97 to associate with a myriad of different cofactors and to form a correspondingly large number of complexes enables its participation in numerous, diverse cellular processes[46,54]. Moreover, the mobility and communication between p97 domains are intimately linked

with the ATPase function, and any hinderance in those domain movements can perturb ATPase activity[55,56].

Over the years, elucidation of the p97 ATPase structure has enabled continuous advances in understanding the regulation of its activity, including visualization of drug-like allosteric and ATP-competitive inhibitors bound to the protein. Optimization of early ATP-competitive inhibitors, DBeQ and ML240[57,58], led to the identification of CB-5083 (1-[4-(benzylamino)-5H,7H,8H-pyrano[3d,4]pyrimidin-2-yl]-2-methyl-1H-indole-4-carboxamide)[59]. CB-5083 was the first reported, highly selective inhibitor acting on the p97 D2 ATPase[59,60], and cancer cells treated with CB-5083 exhibit a remarkable stall in the endoplasmic reticulum-associated degradation (ERAD) process[27,59,61]. However, mutation resistances occurr around CB-5083 binding site, such as N660K and T688A, compromise its efficacy in vitro and presage potential resistance in patients[59,62]. CB-5083 entered Phase I clinical trials in 2014 (NCT02243917 and NCT02223598), but unfortunately unacceptable, albeit reversible, ocular toxicity was observed[63]. A second-generation ATP-competitive inhibitor with re-optimized properties, CB-5339, was developed by Cleave Therapeutics to improve the safety profile and has also entered clinical trials (NCT04372641 and NCT04402541)[64].

A number of other p97 inhibitors have been reported, including competitive as well as allosteric inhibitors[28], such as indole amides[65] and phenyl indoles (such as UPCDC30245)[66,67]. The earliest examples of allosteric p97 inhibitors are represented by an alkylsulfanyl-1,2,4-triazole scaffold that was reported by scientists at Nerviano Medical Sciences and Genentech[68]. Among these, NMS-249 (a covalent inhibitor) and NMS-873 (a non-covalent inhibitor) were investigated using biochemical and cellular tools, demonstrating their tight bindings to wild-type p97 (p97[WT]) and its disease mutants[69]. The triazole allosteric inhibitors have the advantage of being unaffected by the mutants that provide resistance to active-site inhibitors. With uncompromised efficacy, they could be used to treat patients resistant to first-line active-site inhibitors[70,71]. However, the development of NMS-873 was challenging due to its poor solubility and pharmacokinetic properties[68,72]. More recently, newly designed analogs were reported with improved antiproliferative activity ($IC_{50} < 500$ nM) and elevated levels of biomarkers (Ub-K48, CHOP, and cleaved caspase-3), consistent with p97 inhibition at levels similar to the clinical candidate CB-5083[73].

Here, we provide a structural understanding of the allosteric mechanism of inhibition of p97 ATPase by several novel members of this triazole class of inhibitors. Structures of three newly designed inhibitors in complex with p97[WT] or p97[R155H] disease mutant proteins were determined using single-particle cryo-EM, enabling us to acquire high-resolution biomolecular structures of full-length p97 and to analyze the structural heterogeneity of the protein-inhibitor complexes[74]. The structures reveal the power of this method for studying mechanisms of biomolecules in native-like conditions[75]. Our study employs a systematic approach to study the binding of various triazole allosteric inhibitors to wild-type and disease mutant p97 ATPase. Our structures reveal the binding site of triazole inhibitors and the structural changes that lead to the alteration in inter- and intra-domain communication, providing novel insights into their allosteric mechanism of inhibition.

## Results
### Allosteric triazole inhibitors bind to the interface between D1 and D2 domains

Building on prior efforts of p97 drug design based on the triazole scaffold[73], our medicinal chemistry team has generated a series of novel compounds based on the 1,2,4-triazole scaffold containing a cyclohexenyl thioether[73]. Three compounds in this class have been studied here: NSC799462, NSC804515, and NSC819701 (Fig. 1c; Supplementary Data 1–3)[73]. NSC799462 and NSC804515 contain an alkynyl isosteric replacement for one of the biphenyl moieties in NMS-873. NSC819701 contains a 1,2,3-triazole replacement. We performed single-particle cryo-EM of full-length p97 ATPase with these triazole inhibitors to determine their binding modes. The two-dimensional (2D) class averages of p97 |$_{NSC799462}$ showed that

both hexameric and dodecameric forms are present (Supplementary Fig. 1a, b), similar to previous observations on the wild-type or mutant p97[47,62,76–79]. Since we did not find an asymmetric form of the oligomers among our three-dimensional (3D) reconstructions, we enforced symmetrization with C6 and D6 symmetries to the final densities of the hexamer and dodecamer, respectively. The final reconstructions reach resolutions of 3.23 Å and 3.33 Å for hexamer and dodecamer, respectively (Supplementary Figs. 1c, d and 2). The symmetrized cryo-EM densities resolved the side chains of the D1 and D2 domain protein residues and the bound nucleotides and inhibitor compounds (Supplementary Fig. 3).

All three compounds bind at the location previously predicted using cross-linking of the azido analogs of NMS-873 and NMS-249 followed by liquid chromatography-mass spectroscopy (LC-MS) of tryptic peptides[69]. NSC799462 binds at the interface between the D1 and D2 domains of the adjacent p97 monomers (Fig. 2). The inhibitor does not directly interact with bound nucleotide, but its methylpiperidine group approaches the D2 nucleotide-binding site of an adjacent subunit with a distance of approximately 8 Å between the methyl group on the piperidine of NSC799462 and the 3'-hydroxyl group of the adjacent ADP ribose (Fig. 2a). The interactions between NSC799462 and p97 are primarily hydrophobic, involving surrounding residues V493, P496, V497, P500, L504, P510, I531, C535, A537, P571, and F618 (Fig. 2b). K512 and N616 make hydrogen bonds with the pyridine and triazole nitrogens, respectively (Fig. 2b). Q398' of the adjacent D1 domain weakly interacts with the $sp^3$-oxygen atom in the carbamate group of NSC799462 (4.3 Å) (Fig. 2b). The side chain of K615 points away from the binding site and does not engage with the bound compound.

To understand the structural changes resulting from NSC799462 binding, we superimposed our structure with a previous p97 structure (PDB: 5FTK)[30] (RMSD 1.108 Å) (Fig. 2c). In the presence of the inhibitor, a significant structural change in the loop T613-V617 is observed (Fig. 2c, inset), which allows NSC799462 to fit the binding site and enables the interaction of N616 with a triazole nitrogen (Fig. 2b). The width of the binding pocket increases to accommodate NSC799462: the distances between N616 and V497 and between K512 and P496 increase from 3.2 Å and 3.9 Å to 6.7 Å and 7.4 Å, respectively (Fig. 2d). These rearrangements expand the volume of this binding cavity, induce subtle shifts of the surrounding structural motifs, and result in the helices, α2A, α2B, α2C, and α2E moving outwards (Fig. 2c). These spatial rearrangements of the side chains and slight movements of the surrounding helices likely affect p97 motion, thereby inhibiting the ATPase activity.

NMS-873 has a similar structure to NSC799462, but the conformations of the interacting residues differ (Fig. 3a). NSC799462 contains the same triazole scaffold as NMS-873 (Fig. 3a) but has a different substitution pattern on the aryl ring. In addition, NMS-873 has a benzene ring instead of an alkyne, a methyl sulfone instead of a carbamate at the terminus, and a cyclopentyl instead of the asymmetric cyclohexenyl group. The cryo-EM structure of NMS-873 bound to p97 demonstrated that NMS-873 occupies the same binding site as NSC799462 (PDB code: 7LMY) (Fig. 3a)[80]. However, unlike NSC799462, NMS-873 does not interact with N616 and K512, whose side chains point away from the binding site (Fig. 3a). These side chain conformational differences may be due to their interactions with the distinct chemical properties of the two lead compounds.

The previous hypothesis of triazole allosteric inhibition (NMS-249 or NMS-873) was that these compounds alter the conformation of the inter-subunit signaling (ISS) motif, D609-G610, leading to a change in the rotameric conformation of the arginine fingers, thereby blocking the D2 ATP hydrolysis[69,80]. Structural comparison between p97[E578Q] |$_{ATP,NMS-873}$ and p97|$_{ADP,NSC799462}$ demonstrates a closer contact between ISS motif and the arginine finger R638 when bound to NMS-873 and ATP (p97|$_{ADP,NSC799462}$: 3.1 Å; p97|$_{ATP,NMS-873}$: 2.6 Å) (Fig. 3c). The ISS motif and R638 are slightly away from the arginine finger R635 when bound to NMS-873 and ATP (p97|$_{ADP,NSC799462}$: 3.8 Å; p97|$_{ATP,NMS-873}$: 5.4 Å). The NMS-bound p97 structure shows that R635 interacts with γ-phosphate of ATP, but not the mutated Walker B E578Q, while the NSC-799462-bound p97 structure

**Fig. 2 | Cryo-EM structure of p97 with NSC799462 triazole inhibitor.** Surface representation of the two p97 monomers. Green, orange, orange-red, and purple are the N-terminal (NTD), D1, D2, and linker domains in chain A. White (chain B) is one of the adjacent p97 monomers, which interfaces with the binding site of NSC799462. **a** The NSC799462 terminal methylpiperidine group approaches the adjacent D2 nucleotide-binding site at a distance of approximately 8 Å. Hot pink is the D2 domain of the adjacent p97 monomer (D2'). Light blue surfaces are cryo-EM densities for NSC799462 and ADP. **b** NSC799462 binding site. NSC799462 is shown in ball-and-stick. Gray, red, blue, yellow, and light green represent carbon, oxygen, nitrogen, sulfur, and fluorine atoms, respectively. Side chains of the interacting protein residues are shown in sticks. Cryo-EM density is shown in light blue surfaces. **c** Superposition of p97$^{WT}$|$_{NSC799462}$ and p97$^{WT}$ (PDB code: 5FTK) (RMSD 1.098 Å). p97$^{WT}$ structure is shown in semi-transparent white. The region following the C-terminus of the α2 G helix that adopts a different conformation in the presence of inhibitor is shown in the inset. **d** Superposition of p97$^{WT}$ (light purple, without inhibitor) (PDB code: 5FTK) and p97$^{WT}$|$_{NSC799462}$ (orange red, with NSC799462 in surface representation) illustrates the expansion of the binding site to fit the inhibitor. All distances are measured in Å.

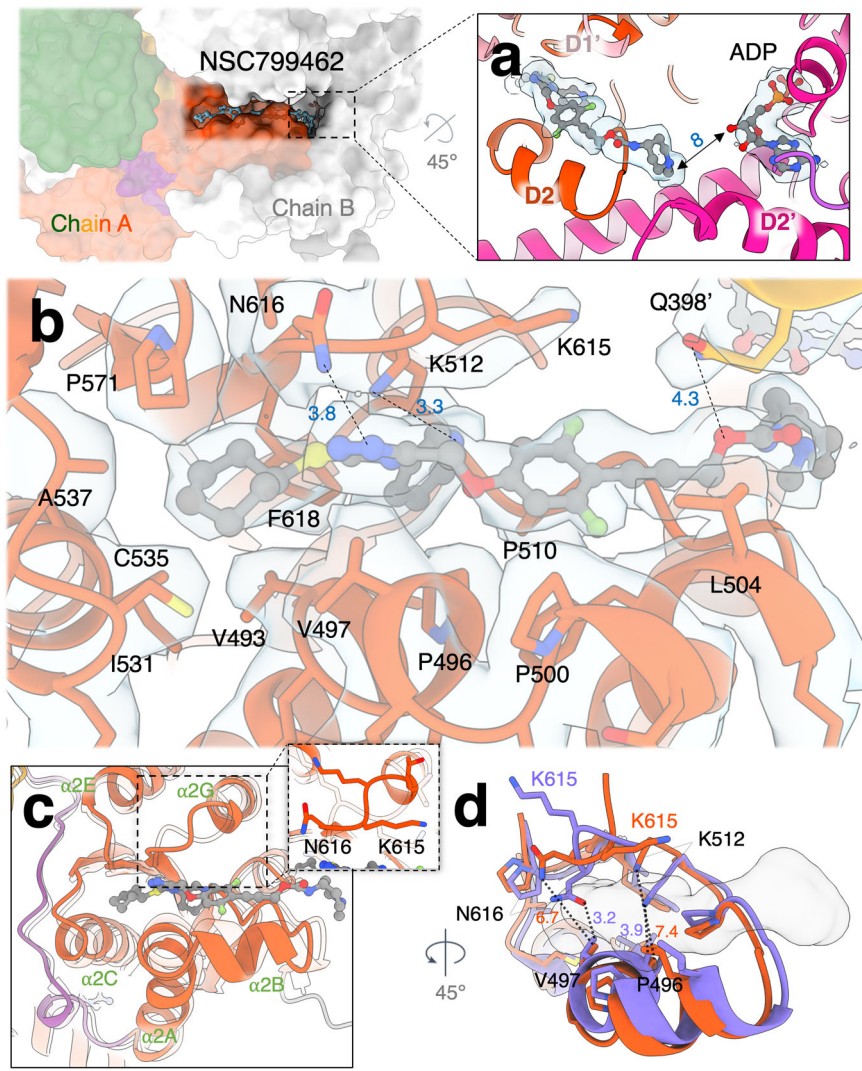

shows that R635 interacts with E578, but away from ADP (Fig. 3c). It is likely that different nucleotide-binding states, such as the ATP-bound state in the previous p97$^{E578Q}$|$_{ATP,NMS-873}$ structure[80], could potentially induce conformational changes in arginine finger R635 and the Walker B motif, resulting in functional modulation of ATPase activity. Therefore, the binding of the triazole inhibitor in various nucleotide states may have different modes of modulations in the interaction between the ISS motif and the two arginine fingers and ATPase activity.

Phenyl indoles, such as UPCDC30245, represent a distinct type of allosteric inhibitors targeting p97 ATPase[70,81,82] and occupy part of the same binding site as NSC799462 (Fig. 3b and Supplementary Fig. 4a)[30]. Superposition of the cryo-EM structures of p97|$_{UPCDC30245}$ and p97|$_{NSC799462}$ reveals a partial overlap between the binding sites of UPCDC30245 and NSC799462 (Fig. 3b and Supplementary Fig. 4b; PDB code: 5FTJ)[30]. Specifically, the 2-phenyl indole of UPCDC30245 occupies the same site as the triazole, pyridine, and cyclohexene rings of NSC799462 (Fig. 3b and Supplementary Fig. 4b). While the residues interacting with the phenyl indole of UPCDC30245 overlap with those interacting with the cyclic moieties of the triazole-based inhibitors, the overall interacting networks of these two inhibitors are distinctly different, leading to a larger buried surface area of 2257.6 Å$^2$ for NSC799462 than that of 1648.9 Å$^2$ for UPCDC30245 (Fig. 3b and Supplementary Fig. 4b). Based on the cryo-EM structure, the binding of UPCDC30245 is mostly stabilized via hydrophobic interactions and lacks specific interactions with surrounding residues (Fig. 3b)[30]. However, an atomic model based on the cryo-EM structure that accounted more fully for

the structural-activity relationship (SAR) in the series, showed hydrogen bonds between the small molecule and Q494, E498, E534, and C535[67]. The p97$^{K512N}$ IC$_{50}$ measurements, in which inhibition by UPCDC30245 is less affected (0.44 µM p97$^{K512N}$ for UPCDC30245) than is inhibition by NSC799462 (0.015 µM p97$^{WT}$ versus 0.045 µM p97$^{K512N}$) is reflective of that area of the binding site not being occupied by the phenyl indoles (Fig. 3d and Supplementary Table S4)[83]. It is important to note that NSC799462 exhibits a lower IC$_{50}$ of 15 nM (Fig. 3d and Supplementary Table S4), indicating a stronger inhibitory potency than that of UPCDC30245, which has an IC$_{50}$ of 27 nM[30]. Therefore, the newly designed triazole inhibitors have a significant impact on p97 activity by critical residues, displaying greater potency than UPCDC30245.

These newly designed allosteric inhibitors demonstrate an enhanced potency compared to NMS-873 in both titrations of p97$^{WT}$ and cellular viability assays (Supplementary Fig. 5 and Table S5–7). The minimal inhibitor concentration that exhibits the maximal inhibition on p97 activity is between 0.07 and 0.22 µM (Supplementary Table S5), closely matching the IC$_{50}$ values for cell viability, which assess the efficacy of these compounds in inhibiting the growth of HCT116 cells (Supplementary Table S6, 7).

## Analysis of the triazole allosteric inhibition in vitro and in cells

We previously generated CB-5083 resistant HCT116 cell lines and demonstrated the antiproliferative effect of CB-5083 is due to inhibition of p97[84]. Here we have established HCT116 colon cancer cell lines that are

**Fig. 3 | Structural and functional comparisons of p97 with various triazole inhibitors. a** NMS-873 binding site (PDB code: 7LMY). Orange-red is the p97 D2 domain. **b** UPCDC30245 binding site (PDB code: 5FTJ). **c** Structural comparison of the nucleotide-binding sites between $p97|_{ADP,NSC799462}$ and $p97^{E578Q}|_{ATP,NMS-873}$. Yellow, blue, and pink sticks are the intersubunit signaling (ISS) motif, arginine fingers, and E578 ($p97|_{ADP,NSC799462}$) or Q578 ($p97^{E578Q}|_{ATP,NMS-873}$). Distances are measured in Å. **d** Histogram comparing $IC_{50}$ measurements (on a logarithmic scale) of triazole inhibitors and CB-5083 on the ATPase activities of various purified p97 mutants ($n = 6$). Data are presented as the mean ± standard deviation (SD). CB-5083 is an ATP-competitive inhibitor. WT represents the wild type.

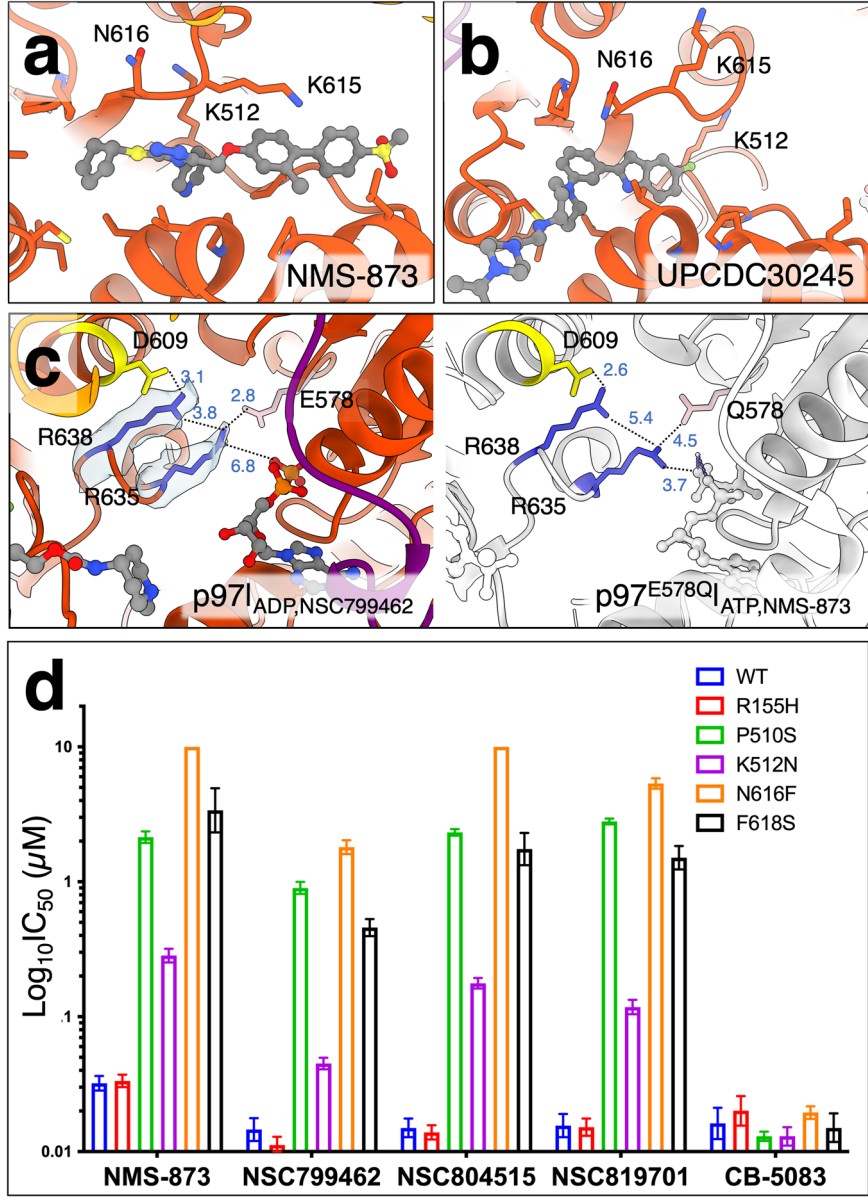

resistant to a related triazole-based allosteric p97 inhibitor, UPCDC30766[73], using the same method used to generate resistance to the CB-5083 p97 active site inhibitor[84]. After isolating single HCT116 clones that survived in the presence of UPCDC30766, we expanded the cells and showed members of the triazole-based allosteric inhibitor class exhibited a 10-fold increase in $IC_{50}$ in cellular viability assays (Supplementary Table S3). We then cloned and sequenced p97 cDNA from the expanded HCT116 clones and identified P510S, K512N, and F618S mutations. These resistance mutations were induced under the selection concentrations that were lethal to most cells, conditions that are not typical in normal cellular environments. These novel mutant p97 proteins, like $p97^{N616F}$, which was shown to be resistant to NMS-873[69], and the R155H disease mutant, were expressed, purified, and assayed for inhibition by NSC799462, NSC804515, NSC819701, NMS-873 and CB-5083 (Fig. 3d). The mutated amino acid residues modify the inhibitory effects of the triazoles in agreement with predictions from the structural results. Note that the K512N mutant was isolated from resistant HCT116 cells in the presence of a racemic mixture of NSC799462, whereas the P510S, K512N, and F618S mutants were isolated from the resistant HCT116 cells in the presence of UPCDC30766. For NSC799462, the $IC_{50}$s of the identified

resistance mutants P510S, K512N, N616F, and F618S exhibited higher values than those of the $p97^{WT}$ with increased $IC_{50}$ values of 61, 3, 121, and 31 fold, respectively (Fig. 3d and Supplementary Table S4). Notably, the $IC_{50}$s of all the allosteric inhibitors were the most significantly increased with $p97^{N616F}$, while the $IC_{50}$ for ATP competitive CB-5083 was only slightly affected (1.2 fold greater) (Fig. 3d and Supplementary Table S4). Therefore, these residues make functionally important interactions with the triazole inhibitors, and substantially contribute to this class of inhibitor's ability to inhibit p97 ATPase activity.

The N616F mutation has the greatest effect on inhibition by the allosteric inhibitors (Fig. 3d). To examine its critical function from a structural perspective, we modeled an in silico mutation of N616F and found that not only does the phenylalanine side chain eliminate the hydrogen bonding between the N616 carboxamide side chain and the triazole nitrogen, but the bulky aromatic ring of the phenylalanine also would sterically hinder binding of the triazole compound (Supplementary Fig. 6). Here we highlight the critical role of the N616 residue, which may not fully capture the impact of the N616F mutation on the overall structural changes in p97. Thus, the presence of a small side chain and the specific

interaction between N616 and the triazole heterocycle appear critical for the high potency of these allosteric inhibitors.

## Triazole inhibitor binding interferes with the domain-domain communication within p97

The overall molecular architecture of ADP-bound p97$^{WT}$|$_{NSC799462}$ is similar to the p97$^{WT}$|$_{ADP}$ structure (PDB code: 5FTK; RMSD 1.098 Å) (Fig. 4a)[30]. The NTD exhibits a down conformation that is stabilized via interactions with the N-D1 linker and D1 domain. However, the N-D1 linkers exhibit distinct conformations in these two structures, leading to different interacting networks between the NTD and D1 domain (Fig. 4a). Since the NTD conformation links to the D1 nucleotide-binding states (both of them are ADP-bound)[30,43], the differential interactions between these two structures are likely due to NSC799462 binding. The acidic residues of the N-D1 linker (192-EDEEE-196) form a hydrogen-bonding network with K63, R93, R95, and K164 of the NTD, stabilizing the NTD in a down conformation (Fig. 4b). The down positions of the NTDs limit their binding to p97 cofactors and thus impede p97's physiological actions[21,45]. Given that the up-NTD conformation was not found in the analysis of the NSC799462-bound p97 cryo-EM data, our results suggest that the NSC799462 binding stabilizes the NTD-down conformation, thus preventing the functional interactions that p97 makes with its cofactors.

Several p97 complexes, including its mutants, have been reported to show a dodecameric form[40,47,62,76,77,85–89]. A higher-ordered dodecamer was also found in the NSC799462-bound p97 complex (Supplementary Fig. 1). As shown in the previous studies[47,78], the dodecamer is formed by the two opposite D2 rings packing against each other via the α2 M helix and part of the C-terminal tail (Supplementary Fig. 7a). Superposition of the monomeric p97 structures of hexamer and dodecamer does not show significant structural differences (RMSD 0.778 Å) (Supplementary Fig. 7b). Although the physiological function of the p97 dodecameric form is still not well understood, it has been suggested to be inactive due to its higher order structure that may limit domain movements[47,76].

## The allosteric triazole inhibitor alters the NTD conformations of the p97$^{R155H}$ mutant

p97 NTD disease mutants show a highly elevated ATPase activity, mainly originating from D2 ATPase[21], possibly due to the disruption of communication between NTD and D1 domain[47,49,53]. Also, the NTD mutation results in NTD exclusively in the up conformation[47] and lower affinity for nucleotides by the ATPase domains[45,47]. To investigate the structure of the R155H NTD mutant with triazole-class inhibitors, we used single-particle cryo-EM to solve the structure of p97$^{R155H}$ proteins with NSC804515 and NSC819701 (Supplementary Figs. 8–12). NSC804515 and NSC819701 share the same pyridyl triazole core and cyclohexenyl thioether but differ in substituents at their termini. Compared to NSC799462, NSC804515 has a 2,5-difluoro substitution pattern on the aryl ring rather than a 3,5-pattern, and a propionamide at its terminus rather than a carbamate, reducing its length (27.4 Å) compared to NSC799462 (30.1 Å) (Fig. 1c). Their molecular lengths were measured after energy minimization. In contrast, NSC819701 contains a significant difference in that the acetylene group in NSC799462 and NSC804515 is replaced with a 1,2,3-triazole, and its terminus is a dimethylmorpholinylethyl amide, resulting in a length (29.1 Å) comparable to NSC799462 (Fig. 1c).

The resolutions of the cryo-EM densities of p97$^{R155H}$|$_{NSC804515}$ and p97$^{R155H}$|$_{NSC819701}$ were sufficiently high to model the coordinates of the lead compounds (Supplementary Fig. 13). As expected, NSC804515 and NSC819701 occupy the same binding site as NSC799462 (Fig. 5a and Supplementary Figs. 13, 14), and the network of interactions with the surrounding protein residues is the same as for NSC799462 (Fig. 5a). Additionally, the terminal carbonyl group of NSC804515 can interact with K663' of the adjacent D2 domain (Fig. 5a). Notably, K615 of the p97$^{R155H}$|$_{NSC804515}$ and p97$^{R155H}$|$_{NSC819701}$ shows different rotameric conformations compared to that of the p97$^{WT}$|$_{NSC799462}$. The K615 side chain allows interaction with 2,5-difluorosubstitued aryl groups in NSC804515 and NSC819701, unlike the K615 side chain observed in the p97$^{WT}$|$_{NSC799462}$ structure, which contains a different difluoro-substitution pattern. (Figs. 1c and 5a). The

**Fig. 4 | Structural superposition of p97$^{WT}$|$_{NSC799462}$ and p97$^{WT}$. a** Superposition of p97$^{WT}$|$_{NSC799462}$ and p97$^{WT}$ (PDB code: 5FTK). Green, orange, orange-red, and purple are the N-terminal (NTD), D1, D2, and linker domains, respectively. p97$^{WT}$ is shown in transparency. **b** Enlarged view of the interaction network between the NTD and N-D1 linker domain. The interaction network is established via hydrogen bonds between protein residue side chains, stabilizing the NTD in the down conformation. All distances are measured in Å. Density fitting is shown in Supplementary Fig. 3b.

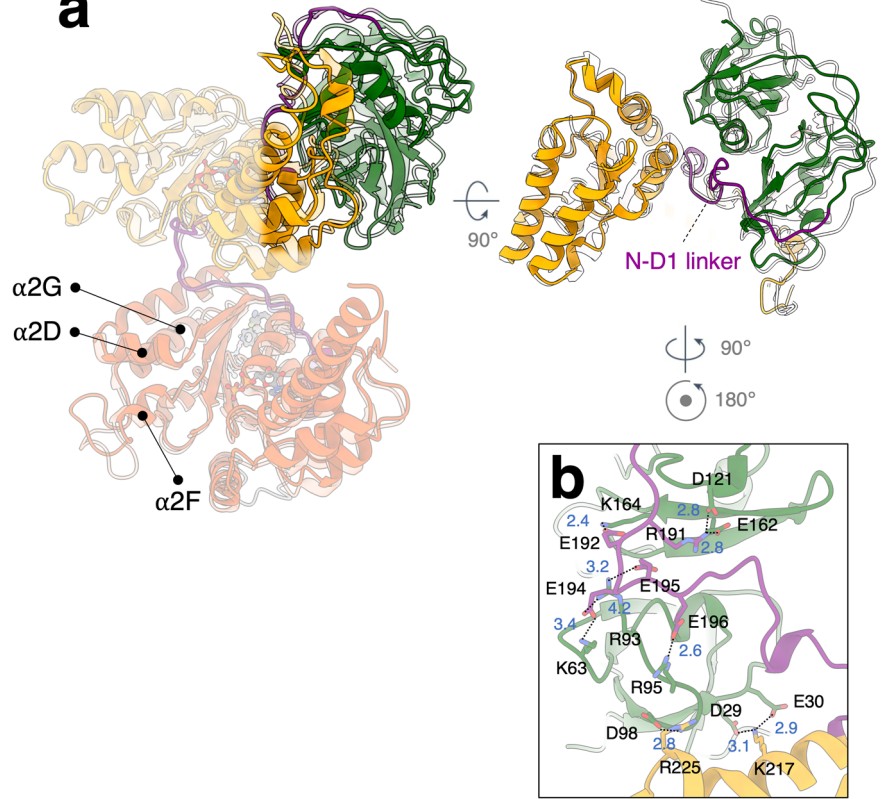

**Fig. 5 | Allosteric inhibition of the p97$^{R155H}$ disease mutant by triazole inhibitors. a** Binding site of the triazole inhibitors. Gold, blue and white are the structures of p97$^{R155H}$|$_{NSC804515}$, p97$^{R155H}$|$_{NSC819701}$, and p97$^{WT}$|$_{NSC799462}$. Yellow, light green, blue, and red ball representations are for sulfur, fluorine, nitrogen, and oxygen atoms. K615 side chain adopts different conformations when interacting with *meta-* and *para-*fluorobenzene rings. **b** Cryo-EM densities of p97$^{R155H}$|$_{NSC804515}$ and p97$^{R155H}$|$_{NSC819701}$. Green, orange, orange-red, and purple are the N-terminal (NTD), D1, D2, and linker domains, respectively. The densities were filtered to 5.0 Å$^{-1}$ for presentation. The NTD shows in different conformations when bound to the triazole inhibitors. Two p97$^{R155H}$|$_{NSC819701}$ structures with different NTD conformations, partially up and down conformations, were determined. **c** Structural superpositions of the p97$^{R155H}$|$_{NSC819701}$ (blue; State B, down NTD) and p97$^{R155H}$|$_{NSC804515}$ (light orange; up NTD) (RMSD 0.846 Å). Dark green is the NTD of p97$^{R155H}$|$_{NSC819701}$. Enlarged view shows the interaction network of the NSC819701 with the surrounding residues of helices α1 J and α1 M. **d** Interaction between NTD and D1 domain of the p97$^{R155H}$|$_{NSC819701}$. Dark green is the NTD of p97$^{R155H}$|$_{NSC819701}$. Gray surfaces are cryo-EM densities of p97$^{R155H}$|$_{NSC819701}$ in NTD-down conformation. All distances are measured in Å.

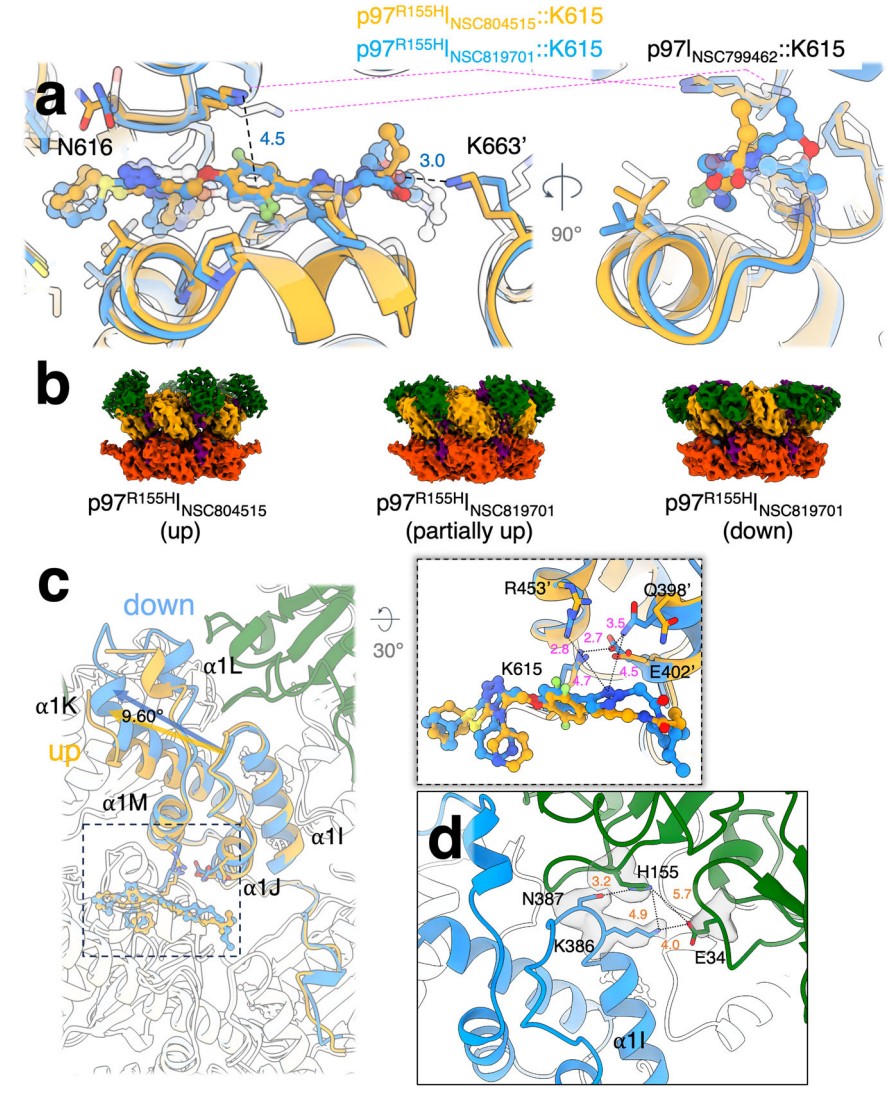

distances between the ε amino group of K615 and the difluorobenzene ring of NSC804515 and NSC819701 are 3.7 Å and 4.2 Å, respectively (Fig. 5a). However, the potencies of NSC799462, NSC804515, and NSC819701 to p97$^{WT}$ are not significantly different (Fig. 3d). Thus, the positions of the fluorine on the benzene ring affect the conformation of the K615 side chain but do not contribute significantly to IC$_{50}$ values of the triazole compounds.

The IC$_{50}$ values for p97$^{WT}$ and p97$^{R155H}$ mutant are also similar for all three triazole compounds, whereas the IC$_{50}$ for K512N is 7.5 to 12-fold higher for NSC819701 and NSC804515, respectively (Fig. 3d). In general, when comparing these mutants to p97$^{WT}$, NSC804515 and NSC819701 were all more than 30-fold resistant to P510S, N616F, and F618S and only 7- to 12-fold resistant to K512N (Fig. 3d). In addition, CB-5083 demonstrated similar IC$_{50}$ values against all six p97 variants. Overall, the changes in P510 and N616 residues have a more deleterious effect on the inhibitory activity of these three triazole compounds.

Functional measurements showed that the IC$_{50}$ of the p97 mutants with NSC819701 lies between those of NSC799462 and NSC804515, except that the p97$^{P510S}$ mutant with NSC819701 has the highest IC$_{50}$ value among the triazole allosteric inhibitors (Fig. 3d and Supplementary Table S4). One possibility could be that the substituted 1,2,3-triazole of NSC819701 has an electron-withdrawing property[90], influencing the π-electron density of the fluorobenzene ring. This chemical property may affect the packing of P510 and the di-fluorobenzene ring and destabilize the π-cation interaction between the fluorobenzene ring and K615 (Fig. 2b). On the other hand, the

terminal dimethylmorpholine ring of the NSC819701 does not seem to interact with surrounding protein residues (Fig. 5a), which may allow some flexibility. Since the bound dimethylmorpholine is located at the D1-D2 intra-domain and D2-D2 inter-domain interfaces, its flexibility of the linked dimethyl morpholine might destabilize or stabilize the interactions between p97 monomers and affect the stability of the entire complex.

Previous studies showed that the NTD of the p97$^{R155H}$ mutant is in the up conformation due to fewer interactions between the NTD and D1 domain[47,49,53]. It was suggested that the NTD R155H mutation disrupts the interaction between the NTD and N-D1 linker or D1 domain, resulting in less stability of the NTD down conformation[47,49,53]. When bound to NSC804515, although the NTD densities were fragmented and did not allow us to build their peptidyl backbones (Supplementary Fig. 8), the NTDs are oriented in the up conformations (Fig. 5b), as shown for other p97 NTD mutant structures[45,47].

Unlike the p97$^{WT}$|$_{NSC799462}$ and p97$^{R155H}$|$_{NSC804515}$ structures, NSC819701 binding to p97$^{R155H}$ introduces two different p97$^{R155H}$ conformations with the NTDs either partially up or down (Fig. 5b). Using the image classification method, we can reconstruct two different conformations of the p97$^{R155H}$|$_{NSC819701}$ structures at 3.60 Å (State A; up NTDs) and 3.42 Å (State B; down NTDs) resolution (Fig. 5b and Supplementary Figs. 10–12). The densities of the down NTDs have a higher resolution than those of the up NTDs, implicating a high mobility of the up NTDs (Supplementary Figs. 10–12).

p97$^{R155H}$|$_{NSC804515}$ retains the fully up NTD conformation (Fig. 5b). To understand how NSC819701 stabilizes the down NTD conformation of p97$^{R155H}$, we superimposed the structures of p97$^{R155H}$|$_{NSC819701}$ (State B) and p97$^{R155H}$|$_{NSC804515}$ (RMSD 1.171 Å) (Fig. 5c). The two compounds have the same specific interactions between N616 and the 1,2,4-triazole and between K615 and the difluorobenzene ring (Fig. 5a). In addition, the substituted 1,2,3-triazole ring of NSC819701 is not coplanar with the adjacent difluorobenzene ring, allowing weak interactions of the N$^1$ of the 1,2,3-triazole with K615 (4.7 Å) and Q398' (4.5 Å) side chains (Fig. 5c). In the partially up NTD forms of the p97$^{R155H}$|$_{NSC819701}$, the 1,2,3-triazole does not show the same conformation that allows an interaction with K615 or Q398' (Supplementary Fig. 13). The new side chain conformations of K615 and Q398' of the NSC819701-bound p97$^{R155H}$ allow the formations of hydrogen bond pairs between the side chains of Q398'-E402 (3.5 Å), K615-E402' (2.7 Å), and K615-R453' (2.8 Å) (Fig. 5c). This interaction network limits the movement between α1 J and α1 M helices. These interactions were not seen between α1 J and α1 M helices in the p97$^{R155H}$|$_{NSC804515}$ structure (Fig. 5c); therefore, they appear to be important determinants of the NTD conformation. Also, the α1K-α1 L helix-turn-helix (HTH) motif interacts with down NTD, but not up NTD[30]. During the motions of the p97 structural motifs, it is possible that the relatively immobilized α1J-α1 M loses the regulatory control of the HTH movement of the α1K-α1 L which can interact with the NTD and stabilize it in the down conformation.

The NTD mutations, including R155H, could disrupt the interactions between the NTD and N-D1 linker and between the NTD and D1 domain, thus favoring the NTD up conformation[53]. The H155 side chain can form a hydrogen bond with N387 and has weak interactions with K386, stabilizing the interaction of NTD with the D1 domain (Fig. 5d). E34 on the NTD also forms a hydrogen bond with the K386 side chain, contributing to its stabilization (Fig. 5d). The interactions of R155 and H155 with their surroundings are slightly different but are formed to stabilize the NTD in the down conformation when the triazole-based inhibitor is present (Fig. 5d). It seems that small structural changes due to the binding of the NSC819701 impact the rotameric conformations of side chains on the nearby helices and rearrange the hydrogen bonding network with the mutated H155 to stabilize the down NTD.

The NTD down conformation has been previously reported in the p97$^{R155H}$|$_{ADP}$ structure[53], whereas the up conformation was observed in previous structural studies in the presence of either ADP or ATPγS[47,49,51]. Superposition of p97$^{R155H}$|$_{NSC719801}$ and p97$^{R155H}$ (PDB code: 7RL6) structures reveals that the NTD adopts a slightly further down conformation when bound to NSC719801, resulting in different interactions of H155 with its surrounding residues, specifically K386 and N387 (Supplementary Fig. 15a, b). The interaction of the R155H with N387 corroborates the previous findings identified by NMR spectroscopy[49]. In addition, structural superposition shows that NSC819701 significantly impacts the loop structure of T613-V617, similar to the observations in the p97|$_{NSC799462}$ structure (Fig. 2c and Supplementary Fig. 15c). Like the effect of NSC799462 binding, the conformational change around the inhibitor binding site affects the conformation of the N-D1 linker and its interactions with NTD or D1 domain.

## Discussion

Here we report structural and functional analyses of both wild-type and R155H mutant human p97 ATPase bound to three allosteric triazole-based inhibitors, providing mechanistic insights into allosteric inhibition of AAA+ ATPase proteins. The allosteric triazole inhibitors presented here demonstrate a better potency compared to the previous generations in both titrations and cellular assays (Supplementary Fig. 6 and Tables 3–7)[30,73]. In addition, modifications to the functional groups of these triazole inhibitors improves their solubility and pharmacokinetics profile without compromising efficacy[73]. In this study, we used single-particle cryo-EM to analyze the reconstructions of triazole-based inhibitor-bound p97 ATPase complexes and assessed the ATPase activities of the mutants to

determine the role of essential binding residues. Obtaining high-resolution structures through crystallizing the p97 complex with allosteric triazole-based inhibitors is distinctly difficult because of its molecular size and flexibility[69]. Single-particle cryo-EM, in contrast, allows access to the ligand binding sites and enables one to capture the protein's structural heterogeneity in its solution states. Our results showed the heterogeneous nature of the triazole-bound p97 ATPase, particularly the two different NTD conformations of the p97$^{R155H}$|$_{NSC819701}$ and two different oligomerization states of the p97|$_{NSC799462}$. Additionally, we conducted a cell-based screen for resistance mutants that identified changes to four amino acid residues in p97 that line the binding site for these allosteric inhibitors. Each of these mutants was purified and characterized with five different inhibitors (three triazole inhibitors, NMS-873, and CB-5083) to determine the critical role of each residue for inhibitor binding. Although there is considerable data on the structure and function of p97 with CB-5083 and NMS-873, the interactions of various triazole allosteric inhibitors with p97 and its disease mutant have not been thoroughly examined in previous studies. Also, the mutational landscapes of p97 significantly differ between normal and cancer cells, affecting the potency of the allosteric inhibitor[69]. This underscores the importance of understanding the interactions between the inhibitor and p97 disease mutants. By identifying how lead optimization to the functional groups of inhibitors influences their binding to wild-type and mutant p97, we can enable the design of targeted drugs with enhanced specificity, particularly for applications in oncology.

The structure and function of p97 ATPase are tightly regulated by its nucleotide states[29]. Also, a previous study of NMS-873 with p97 variants indicated that NMS-873 preferentially binds to p97 in its nucleotide pre-bound states, potentially hindering ATP hydrolysis or preventing ADP release[91]. Our investigation initially focused on the inhibitor binding of p97 in ADP-bound state for the following reasons: (1) The nucleotide-binding pockets of p97 are predominantly occupied by ADP rather than ATP, particularly in D1 domain[92–95]. For p97 ATPase to become active, the bound ADP must leave the nucleotide-binding pocket to allow the entry of ATP. Therefore, the ADP-bound state represents a closer approximation to the ground state of the enzyme and likely constitute the predominant population in a cellular context. (2) Cancer cells generally exhibit a reduced ATP/ADP ratio due to enhanced aerobic glycolysis[96], the so-called Warburg effect, suggesting that the ADP-bound state could be a more effective target of triazole inhibitors as an anti-cancer drug. Therefore, studying the binding of the triazole inhibitors to the ADP-bound state of p97 allows us to assess their interactions in an environment more akin to the cellular milieu in normal or disease conditions, such as cancer.

In addition to our p97|$_{NSC799462}$ structure, the dodecameric form has been reported when it is bound to CB-5083 and an NTD mutant, such as R155H[47,62]. However, the functional implication of this higher-order organization is still not clear[47,62]. The dodecameric form has been suggested to be a heightened state for engaging with substrates[76] or a storage form[79], in which this higher-ordered structure is disassembled in the presence of nucleotides and prepared to function[47,79].

The NTD conformation affects the binding of p97 cofactors, such as p47 or p37, that are essential for maintaining normal p97 function and control of cellular activities[45]. It has been suggested that the down NTDs have limited access to p97 cofactors, reducing cellular activities of p97 ATPase. The down NTD is linked to the ADP-bound state of the D1 ATPase domain[30,45]. However, the NTD of the p97$^{R155H}$ mutant is found to always be in the up conformation[45]. To the best of our knowledge, our p97$^{R155H}$|$_{NSC819701}$ structure is the first structure showing the p97$^{R155H}$ mutant with NTDs in the down conformation. The mechanism for producing this novel NTD conformation in the R155H mutant is likely a response to binding the difluorobenzene ring and terminal dimethylmorpholinyl group of the inhibitor, thereby modulating the interactions

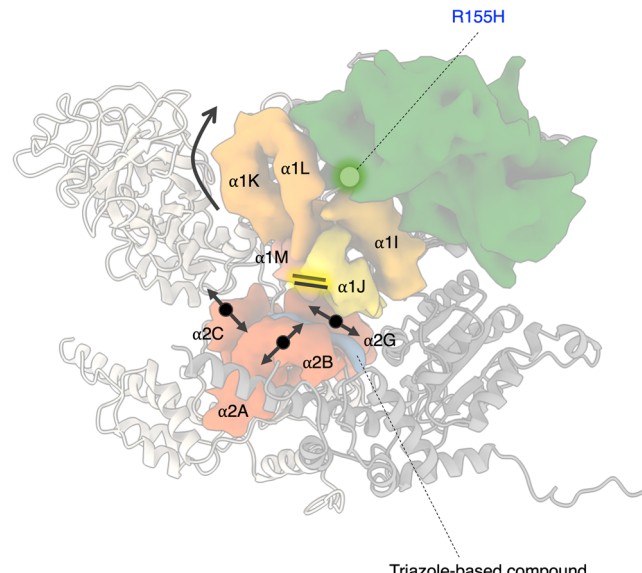

**Fig. 6 | Proposed mechanistic model of the triazole allosteric inhibition on p97 ATPase.** Two adjacent p97 monomers are shown in gray cartoons. Surface representations are the area impacted by the triazole inhibitor binding, which are the D1 domain and the adjacent D2 domain. Dark green is the N-terminal domain (NTD), orange, yellow, and coral are helices of D1 domain, orange-red are helices of D2 domain, and blue is the triazole inhibitor. R155H position is highlighted in the NTD.

between local helices and limiting their motions and rearranging the interactions between surrounding residues. These domain movements influence the interactions in its vicinity, leaving the NTD in either an up or down conformation.

The mutation resistances induced by these triazole inhibitors do not overlap with those induced by ATP-competitive inhibitors, allowing these allosteric inhibitors to be potentially effective in patients who have developed resistance to active-site inhibitors[70,71]. Resistance mutations frequently emerge during targeted therapy, and the development of inhibitors with multiple modes of action is required to effectively address this challenge[70,97]. These improvements make them promising candidates to overcome resistance associated with first-generation drugs, such as CB-5339.

To enhance the pharmacokinetic characteristics and solubility of inhibitor, like NMS-873, of the previous generation, a recent medicinal chemistry study delved into optimizing the inhibitor design[73]. The newly developed triazole inhibitors used in this study exhibit higher solubility and improved pharmacokinetic properties compared to NMS-873[73]. Furthermore, the study demonstrated the enhanced potency of these candidate compounds through cellular assays[73]. Our structural results contribute to a deeper understanding of the SAR associated with these novel triazole lead compounds. Cryo-EM offers direct visualization of the interaction between the lead compound and the target protein, yielding valuable insights for rational drug design.

In summary, we propose a mechanistic model for the allosteric inhibition mode of p97 ATPase and its disease-causing R155H mutant by triazole-based inhibitors. The triazole inhibitors bind between the D1 and D2 domains and at the interface with an adjacent monomer. Inhibitor binding leads to a structural change in the loop at the C-terminus of the α2 G helix (Figs. 2b, c and 6 and Supplementary Movie 1) that is close to the D2 ring region, likely modulating the D2 ATPase activity. Different substitutions on the 1,2,4-triazole scaffold can modulate interactions between helices of the D1 domain (Fig. 6). The terminal group of NSC804515 can interact with the adjacent D2 domain, whereas the 1,2,3-triazole on NSC819701 can interact with the residues on the helices α1 M and α1 J limiting their relative motion and enforcing the NTD down

conformation in the R155H mutant (Fig. 6 and Supplementary Movie 2). We also identified how the side chain in disease mutant, R155H, can also rearrange with surrounding residues of α2I and α2 L to form a hydrogen bonding network and stabilize the interactions between NTD and D1 domains.

## Conclusion
Human p97 ATPase is a significant drug target for neurodegenerative disorders and cancers. This study investigates the structure-function relationships between examples of triazole-based allosteric inhibitors and p97, including the disease mutant R155H and a series of variants identified in a screen for resistant mutants. These findings offer valuable insights into the mechanism of allosteric inhibition of ATPase activities by these triazole-containing inhibitors. Through a systematic examination of the interactions of these lead inhibitors with p97, we have enhanced the opportunities for the design and understanding of small molecule inhibitors based on the triazole scaffold. This structural and mechanistic knowledge can be leveraged for the rational design of new drugs and medical interventions for p97-related critical diseases, such as IBMPFD/ALS or cancer.

## Methods
### Protein purification
Plasmids generated for wild-type and mutant p97 proteins can be found in Supplementary Table S1. Proteins used in this study were purified as previously described[98].

### Synthesis of triazole compounds
Schemes of organic synthesis were described in LaPorte et al.[73]. Characterization of the synthesized lead compounds, including nuclear magnetic resonance (NMR) spectroscopy data, can be found in Supplementary Data 1–3.

### Generation of triazole compound-resistant cell lines
Drug-resistant HCT116 cell lines were derived using the method previously described and with certain modifications[70,99]. Briefly, $10^5$ cells were plated into a 10-cm plate and cultured in the presence of 1.25 μM or 2.5 μM of compounds until drug-resistant clones were visible. Drug-resistant clones were transferred into a new plate for passaging, and subsequently used to study drug resistance and identify the p97 mutations.

### RNA extraction, molecular cloning, and sequencing
Cloning of p97 from the resistant clones was described previously[70]. Purified plasmids were sequenced by Sanger sequencing. For PCR and sequencing primers, refer to Supplementary Table S2.

### ATPase activity measurements
To compare the ATPase activity of wild-type and mutant p97 proteins, each purified protein was diluted to a final monomer concentration of 25 nM in 50 μL ATPase assay buffer (50 mM Tris-HCl (pH 7.4), 20 mM $MgCl_2$, 1 mM EDTA, 0.5 mM TCEP, 0.01% Triton X-100, and 80 nM BSA) containing 200 μM ATP. After 60-min incubation at room temperature, 50 μL Biomol Green reagent (Enzo Life Sciences, Farmingdale, NY) was added to stop the reaction, and the absorbance at 635 nm was measured using a BioTek Synergy Neo 2 plate reader (Agilent, Santa Clara, CA). To determine $IC_{50}$ values of p97 inhibitors, 8 point, 3-fold dilution series compounds were added to the reaction covering a range of concentrations from 30 nM to 6.6 μM or 3 nM to 0.66 μM. Results were calculated from 6 replicates using GraphPad Prism 7.0 (GraphPad Software, Boston, MA).

### Cellular Ub$^{G76V}$-GFP accumulation assay
Ub$^{G76V}$-GFP/ODD-Luc HeLa cells[98] were plated as 4500 cells per well per 30 μL in 384 well plate (Greiner 781091). Clear DMEM containing 2.5% FBS, 1% L-glutamine, and 1% penicillin-streptomycin was used as an assay medium. The next day, cells were treated with the compounds (serial 2-fold dilutions with a starting concentration of 42, 10.5, or 4.2 μM, eight

concentrations) and incubated for 6 h. ImageXpress high-content confocal microscope (Molecular Devices, LLC., San Jose, CA) was used to image cells at GFP channel. The percentage of Ub$^{G76V}$-GFP accumulation (%Acc) is calculated using the following formula:

%Acc = (GFP signal of test compound – GFP signal of DMSO)/(GFP signal of 5 μM NMS-874 – GFP signal of DMSO) * 100.

The IC$_{50}$ values were calculated by fitting the %Acc from four replicates with various concentrations of compounds using GraphPad Prism 7.0 (GraphPad, Boston, MA).

## HCT116 proliferation assay

Anti-proliferative activity was measured using CellTiter Glo® Luminescent Cell Viability Assay (Promega G7572) according to the manufacturer's procedure as described[57]. RMPI1640 containing 5% FBS and 1% penicillin-streptomycin was used as a cell viability medium. A standard curve of HCT116 was generated for finding the linear relationship between the relative luminescence unit and the number of viable cells. Generally, 30 μL of cell suspension was plated in a 384-well plate (Greiner 781080) with serial 2-fold dilutions (from 30,000 to 284 cells per well). 24 h after seeding, 8 μL of media containing 5% DMSO was added to each well, and the plate was incubated for an additional 48 h. To test the anti-proliferative activity of p97 inhibitors, we seeded HCT116 at 750 cells per well according to the linear range. 24 h after seeding, cells were treated with the compounds (serial 3-fold dilutions with a starting concentration of 10.5 μM, eight concentrations). After 48 h of treatment, cell viability was measured by adding Cell-Titer Glo reagent and reading the signal with a BioTek Synergy Neo plate reader (Agilent, Santa Clara, CA). The IC$_{50}$ values were calculated from four replicates using the percentage of growth of treated cells versus the DMSO control. The results were analyzed using GraphPad Prism 7.0 (GraphPad, Boston, MA).

## Cryo-EM data collection

Purified p97 protein was mixed with each triazole-based inhibitor at a molar ratio of 1:50 (p97 protein: compound) in the presence of 1 mM ADP. The mixture was incubated at 24 °C for 1 h. A C-flat holey-carbon coated copper grid (2/1 4 C; Protochips, Morrisville, NC) used for cryo-EM imaging was glow-discharged with a current of 15 mA for 15 s in a Pelco easiGlow glow-discharge system (Ted Pella, Redding, CA). Both sides of the carbon film were glow-discharged. 6 μl of 0.25 mg/mL protein sample was applied on the EM grid, and excess solution was removed by blotting with a filter paper before quickly freezing in liquid ethane. The plunge freezing process was automated using a Vitrobot Mark IV plunge freezer (Thermo Fisher/FEI, Hillsborough, OR) at a humidity of 90% with a blotting time of 5 s.

All cryo-EM data collections were completed in the Eyring Materials Center (EMC) at Arizona State University (ASU). The grid specimen was imaged using a Thermo Fisher/FEI Titan Krios transmission electron microscope (TEM) (Thermo Fisher/FEI, Hillsborough, OR) at an accelerating voltage of 300 keV. Electron scattering was recorded by a Gatan Summit K2 direct electron detector (DED) camera (Gatan, Pleasanton, CA). Nominal magnification was set to 48,077×, corresponding to a physical pixel size of 1.04 Å/pixel at the specimen level. The defocus was set to vary from −0.8 to −2.5 μm, and the camera counted rate was calibrated to 8 e⁻/pixel/second. Exposure time was 8 s, accumulating a total dosage of 47.84 e⁻/Å². The p97$^{R155H}$|$_{NSC804515}$ dataset was collected in super-resolution mode, and the other two datasets, p97$^{R155H}$|$_{NSC819701}$ and p97|$_{NSC799462}$, were collected in counting mode. The beam-image shift scheme was applied to accelerate data collection[100]. The procedure of low-dose imaging was automated using SerialEM software (version 3.8)[101] with customized macros. Details of the cryo-EM imaging conditions are summarized in Supplementary Table S8.

## Image processing

Image processing was generally conducted using cryoSPARC (version 2.14.2)[102] or Relion software (version 3.1-beta-commit-ca101f)[103]. For the p97|$_{NSC799462}$ and p97$^{R155H}$|$_{NSC819701}$ datasets, 3345 and 4355 cryo-EM movies were imported into cryoSPARC for data processing. Motion correction was performed using the 'Patch motion correction', and the defocus parameters were estimated using the 'Patch CTF estimation'. After image curation, particle selection, and 2D classification, 168,754 and 128,935 particles were initially used for ab initio 3D map generation for p97|$_{NSC799462}$ and p97$^{R155H}$|$_{NSC819701}$, respectively. Final particle selection was further conducted using iterative 2D and 3D classification procedures. The ab initio maps were calculated using the stochastic gradient descent (SGD) method[102]. Maps were then refined against the experimental particle images with a C6 symmetry imposed for p97|$_{NSC799462}$ and p97$^{R155H}$|$_{NSC819701}$ hexamers and a D6 symmetry imposed for the p97|$_{NSC799462}$ dodecamer. Resolutions were estimated using the golden standard Fourier-shell correlation (FSC) method at a cutoff of 0.143[104]. The resolutions of the 3D reconstructions for the hexamer and dodecamer of the p97|$_{NSC799462}$ and State A (up NTD) and B (down NTD) of the p97$^{R155H}$|$_{NSC819701}$ are 3.23 Å, 3.33 Å, 3.60 Å, and 3.42 Å, respectively. Local resolution estimation was followed using the previous method implemented in the cryoSPARC program (Supplementary Fig. 16)[105].

For the p97$^{R155H}$|$_{NSC804515}$ dataset, 1,711 movies were used for single-particle image processing. Specimen movements between image frames were translationally registered and averaged using Relion software[103]. Frame averages were dose-weighted and 2 × Fourier-cropped at a spatial frequency of 1.04 Å⁻¹. The defocus and astigmatism of individual images was estimated using the CTFFIND program (version 4.1.13)[106]. 97,583 particle images were automatically selected from the images using a Gaussian blob as a template. Classes with discernible features (27,998 particles) were selected for ab initio volume generation[102]. Initial density was refined against the experimental images with C6 symmetry imposed using regularized likelihood optimization in Relion. The refined map was further improved using 'CtfRefine' and Bayesian polishing procedures to a resolution of 3.31 Å. The selected particle stack and the refined map were then carried over to cisTEM software for refinement with projection matching (version 1.0.0)[107]. Resolutions of the final reconstructions at 3.22 Å were calculated using golden standard FSC criteria at 0.143 cutoff[104]. Local resolutions were estimated using Resmap software (version 1.1.4)[108]. Details of the processing statistics are provided in Supplementary Figs. 1, 7, 8 and Supplementary Table S8.

## Modeling

The previous atomic coordinates of the p97 protein (PDB code: 5FTK and 5FTN)[30] were used as starting models. The initial coordinate was first docked into individual cryo-EM density using the 'Fit in the Map' function in UCSF Chimera software (version 1.14)[109]. The fitted model was rebuilt using Coot (version 0.9-pre)[110]. The densities for N-terminal domains (NTD) of p97$^{R155H}$|$_{NSC804515}$ and p97$^{R155H}$|$_{NSC819701}$ (State A) were fragmented, which did not allow us to model atomic coordinates of NTD. Structures of the allosteric inhibitors, NSC799462, NSC804515, and NSC819701, were built in Coot, and their topologies were optimized using an AM1 geometry in the eLBOW program (version 1.18.2-3874)[111]. The rebuilt coordinates of the p97 and the inhibitor were refined against the corresponding cryo-EM density using the 'phenix.real_space_refine' program in the Phenix software package (version 1.20.1-4487)[112]. Model refinement statistics are listed in Supplementary Table S8. The cross-correlations between the map and model per residue are shown in Supplementary Fig. 18. The Q-scores of modeled residues of the binding site were calculated[113] and shown in Supplementary Fig. 19. The molecular graphic presentation for the final models was made using UCSF Chimera or UCSF ChimeraX (version 0.91)[114].

Structural morphing was used to illustrate the transitions between various conformations from triazole inhibitor binding. The movies were generated using morphing function in UCSF ChimeraX[114]. Supplementary Movie 1 was generated using the morphing from the unbound state (PDB code: 5FTK)[30] to the bound state (p97|$_{NSC799462}$). Supplementary Movie 2 was generated from the p97$^{R155H}$ structures with the NTD in the down conformation (p97$^{R155H}$|$_{NSC819701}$, down NTD) to various up conformations (p97$^{R155H}$|$_{NSC819701}$, partial-up NTD; p97$^{R155H}$|$_{NSC804515}$, up NTD).

**Reporting summary**

Further information on research design is available in the Nature Portfolio Reporting Summary linked to this article.

**Data availability**

Cryo-EM density maps (MRC format) reconstructed in this study were deposited in the Electron Microscopy Data Bank (EMDB) under accession numbers EMD-42603 (p97 |$_{NSC799462}$ hexamer), EMD-44748 (p97 |$_{NSC799462}$ dodecamer), EMD-42625 (p97$^{R155H}$ |$_{NSC804515}$), EMD-42626 (p97$^{R155H}$ |$_{NSC819701}$, State A), and EMD-42627 (p97$^{R155H}$ |$_{NSC819701}$, State B). Model coordinates were deposited in the Worldwide Protein Data Bank (wwPDB) under accession numbers 8UV2 (p97 |$_{NSC799462}$ hexamer), 9BOQ (p97 |$_{NSC799462}$ dodecamer), 8UVO (p97$^{R155H}$ |$_{NSC804515}$), 8UVP (p97$^{R155H}$ |$_{NSC819701}$, State A), and 8UVQ (p97$^{R155H}$ |$_{NSC819701}$, State B). All the data are available in the EMDB and wwPDB databases or from the corresponding author upon request.

**Abbreviations**

| | |
|---|---|
| AAA+ | ATPase associated with diverse cellular activities |
| VCP | valosin-containing protein |
| ER | endoplasmic reticulum |
| IBMPFD | inclusion body myopathy associated with Paget disease of bone and frontotemporal dementia |
| UPS | ubiquitin-proteasome system |
| ALS | amyotrophic lateral sclerosis |
| NMR | nuclear magnetic resonance |
| cryo-EM | cryogenic electron microscopy |
| ERAD | endoplasmic reticulum-associated degradation |
| 2D | two-dimensional |
| 3D | three-dimensional |
| ISS | inter-subunit signaling |

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

## Acknowledgements

The project was supported in part by US federal funds from the National Cancer Institute (NCI), the National Institutes of Health (NIH), under Contract No. 75N91019D00024, through the NExT Chemical Biology Consortium (CBC). We thank Dewight Williams and David Lowry for assisting with the electron microscopy experiments conducted at the Eyring Materials Center (EMC) at Arizona State University (ASU). We acknowledge the utilization of the Titan Krios transmission electron microscope (TEM) supported by NSF MRI 1531991 funding for the instrumentation at ASU EMC. We thank Andre White (Xtal BioStructures, Natick, MA) for providing guidance in building the molecular topology of the triazole compounds. We thank the GPU device support by the NVIDIA GPU Grant Program to P.-L.C.

## Author contributions

F.W. isolated triazole compounds from resistant HCT116 cells, cloned the p97 gene from all resistant single clones, measured ATPase activity, and analyzed the resulting data. R.C.A.C. purified the p97 ATPase and its mutants for cryo-EM structural analysis. T.G. cloned the p97 gene from the resistant single clones and purified the corresponding resistant p97 proteins.

S.L. purified p97 ATPase and its mutants and measured their ATPase activities. P.N. and Y.-P.P. prepared samples for cryo-EM imaging. K.D. and P.-L.C. analyzed cryo-EM image data and built atomic coordinates. P.W., M.G.L., T.T.T., J.D.W., H.S., J.M., W.P., M.W., and D.M.H. conceptualized and synthesized the novel triazole analogs. F.W., K.D., P.N., Y.-P.P., and P.-L.C. prepared the figures. K.D., P.N., P.W., M.G.L., A.J.F., D.M.H., T.-F.C., and P.-L.C. prepared, wrote, and edited the manuscript. T.-F.C. and P.-L.C. conceptualized the experimental design and supervised the structural and biological research studies.

## Competing interests

The authors declare no competing interests.
