## [Peer Review File · Communications Chemistry]

This manuscript has been previously reviewed at another Nature Portfolio journal. This document only contains reviewer comments and rebuttal letters for versions considered at Communications Biology

REVIEWERS' COMMENTS:

Reviewer #1 (Remarks to the Author):

I am satisfied with the revised manuscript to be published in Communication Biology.

Reviewer #2 (Remarks to the Author):

The manuscript describes the structure and function of previously reported triazole inhibitors for the p97 ATPase, examining both wild-type and disease mutant forms of p97.

The major concern is still the lack of novelty given the existing structures of p97 and complexes, including with earlier generations of triazole allosteric inhibitors (cited in the manuscript). This diminishes the value of one of the major claims of the paper, the exploration of the mechanisms of such inhibitors. The cryo-EM structures of the three inhibitor complexes nonetheless contribute incremental knowledge and are of interest to the wider community targeting p97 with allosteric inhibitors. The authors also provide complex structures of the inhibitors with the disease R155H p97 mutant, which advance our understanding of the stabilization of “up” and “down” conformations and possibly of mode of inhibition of the disease mutants.

A large portion of the results are supported by supplementary figures. 19 supplementary figures seem excessive in a manuscript with 6 figures, it will be very difficult for the reader to keep track.

As pointed out before the “Results” section of the paper is very difficult to follow. The authors require more specific examples of this, but that would be a huge undertaking for a reviewer. A few examples follow:

1st paragraph (lines 141-149): additional information on the p97 domains and relevant residues is introduced, not mentioned in the introduction, and with widely insufficient details for a non-expert in the field.

165-176: interactions are described in detail, but no relevance is added

187-195 describes differences with NMS-873 attributed to “distinct chemical properties”, although the interactions described are for a part of the molecule which is identical, triazole N and pyridine N with K616 and K512 (conclusions not substantiated?)

196-209: the relevance is dubious given the difference in bound nucleotide?

210: should this be a new sub-heading? Cell-assay results

233-241 effect of the N616F mutation. This does not seem to be in agreement with the similar NMS-873 structure?

242: UPCDC30245 was not previously introduced

Line 293: this section is very difficult to follow given how the different structures are introduced and compared multiple times, and the intercalating of structure, IC50, mutants (including P510S, its relevance here is not clear). Different complexes, mutants and up and down conformation alternate in a difficult to follow pattern. Twice NSC719801 is mentioned, which is very confusing to the reader. Some comparisons seem less meaningful as 368-380 where two structures in which two variables are changed (mutant AND ligand) are compared.

Line 387 which findings from NMR spectroscopy?

Figure 3c: relevant residues should be labeled

RESPONSES TO THE REVIEWERS

We very much appreciate reviewers' time and comments.

Reviewer #1 (Remarks to the Author):

I am satisfied with the revised manuscript to be published in *Communication Biology*.

We thank Reviewer #1 very much for Reviewer #1's time and help to improve our manuscript.

Reviewer #2 (Remarks to the Author):

The manuscript describes the structure and function of previously reported triazole inhibitors for the p97 ATPase, examining both wild-type and disease mutant forms of p97.

The major concern is still the lack of novelty given the existing structures of p97 and complexes, including with earlier generations of triazole allosteric inhibitors (cited in the manuscript). This diminishes the value of one of the major claims of the paper, the exploration of the mechanisms of such inhibitors. The cryo-EM structures of the three inhibitor complexes nonetheless contribute incremental knowledge and are of interest to the wider community targeting p97 with allosteric inhibitors. The authors also provide complex structures of the inhibitor with the disease R155H p97 mutant, which advance our understanding of the stabilization of "up" and "down" conformations and possibly of mode of inhibition of the disease mutants.

A large portion of the results are supported by supplementary figures. 19 supplementary figures seem excessive in a manuscript with 6 figures, it will be very difficult for the reader to keep track.

We sincerely appreciate Reviewer 2 for the valuable suggestions and comments, which have significantly enhanced the quality of our manuscript. We agree that readability is critical for effectively presenting results and facilitating communication with the scientific community. Furthermore, we recognize that the integrity and completeness of the work are essential for providing readers with clarity and transparency in our data with their analyses. We are immensely grateful for the time and effort invested by the reviewers, and their contributions have enriched our work with their expertise and perspectives.

As pointed out before the "Results" section of the paper is very difficult to follow. The authors require more specific examples of this, but that would be a huge undertaking for a reviewer. A few examples follow:

1st paragraph (lines 141-149): additional information on the p97 domains and relevant residues is introduced, not mentioned in the introduction, and with widely insufficient details for a non-expert in the field.

We appreciate Reviewer 2 for the comments and move this paragraph to the Introduction (Page 5, Line 103).

165-176: interactions are described in detail, but no relevance is added

The interactions between the triazole allosteric inhibitor and its surrounding residues were listed and analyzed to pinpointing their critical spatial arrangements. The impact of these interactions on structure and function was assessed through the subsequent structural comparisons (such as a local loop conformational change, rotameric arrangements of side chains of surrounding residues in the presence of various allosteric inhibitors, and complex structures of p97 disease mutants) and cell assays (such as in-cell EC₅₀ and IC₅₀ measurements and ATPase activity measurements).

187-195: describes differences with NMS-873 attributed to “distinct chemical properties”, although the interactions described are for a part of the molecule which is identical, triazole N and pyridine N with N616 and K512 (conclusions not substantiated?)

Among the three triazole inhibitors (resulting in five structures) investigated in this study, N616 and K512 consistently exhibited the same binding mode (**Supplementary Fig. 13**). When comparing the structure of the p97^{E578Q}_{|ATP,NMS-873}, we hypothesize that the alternate conformation of N616 may be due to the different chemical properties of the two lead compounds, resulting in conformational changes in the interacting residues.

The structures of the two triazole inhibitors, NSC799462 and NMS-873, differ in their alkyl rings attached to the sulfur and their overall dimensions in length (**Fig. 1c**). The ring strains between a cyclopentane and a cyclohexene may affect its mobility and conformations of the surrounding residues. The conformational change of the P571 side chain may be influenced by this (**Fig. R1**), which could similarly impact the N616 conformation. Secondly, the length of the triazole inhibitor can impact the local structural change. Since NSC799462 is longer than NMS-873, it has a larger buried area (2,258 Å²), potentially necessitating a rearrangement of the side chain rotameric conformations to accommodate the inhibitor.

Fig. R1 | Structural superposition of p97^{WT}_[ADP,NSC799462] and p97^{E578Q}_[ATP,NMS-873] (PDB code: 7LMY). Orange red cartoon and sticks are p97^{WT}_[ADP,NSC799462] and white for p97^{E578Q}_[ATP,NMS-873]. Triazole inhibitors are presented in a ball-and-stick representation. Green, yellow, blue, and red atoms are fluorine, sulfur, nitrogen, and oxygen, respectively.

196-209: the relevance is dubious given the difference in bound nucleotide?

We agree that the bound nucleotide modulates the triazole allosteric inhibition in part, but not as a whole. The structural comparison between p97_[ADP,NSC799462] and p97_[ATP,NMS-873] demonstrates the likelihood of the functional modulation through the bound nucleotide and triazole inhibitor as well as the inter-subunit

signaling (ISS) motif (D609-G610) and arginine fingers (R635 and R638).

210: should this be a new sub-heading? Cell-assay results

We agree with Reviewer 2's comment, and we add a heading to create a new subsection.

(Page 8, Line 213) Subsection heading: "*Analysis of the triazole allosteric inhibition in vitro and in cells*"

To improve the flow, we move the section for the structural comparison for p97_{UPCDC30245} (Page 8, Line 212) close to those for p97^{E578Q}_{NMS-783}.

233-241 effect of the N616F mutation. This does not seem to be in agreement with the similar NMS-873 structure?

The ATPase activity measurements indicated the N616F mutation exhibited the highest IC₅₀ response in the presence of NMS-873, NSC799462, NSC804515, or NSC819701 (**Fig. 3d**), suggesting that N616F has the most significant impact under triazole allosteric inhibition compared to other mutations at the relevant residues. The p97 structures bound with the three various triazole allosteric inhibitors investigated in this study were all demonstrated a consistent binding mode. Specifically, the interaction between the N616 and the triazole nitrogen was observed across all structures. This suggests a critical role for the N616 in mediating the effects of triazole allosteric inhibitors on p97.

242: UPCDC30245 was not previously introduced

UPCDC30245 is a phenyl indole class of p97 allosteric inhibitor, and their complex structure in ADP-bound state has been published in Banerjee *et al.* (2016). To improve its introduction, we add UPCDC30245 in the Introduction:

(Page 6, Line 127) From "*A number of other p97 inhibitors have been reported, including competitive as well as allosteric inhibitors²⁸, such as indole amides⁶² and phenyl indoles.^{63,64} The earliest examples ...*"

To "*A number of other p97 inhibitors have been reported, including competitive as well as allosteric inhibitors²⁸, such as indole amides⁶² and phenyl indoles (such as UPCDC30245).^{63,64} The earliest examples ...*"

Banerjee, S. *et al.* 2.3 Å resolution cryo-EM structure of human p97 and mechanism of allosteric inhibition. *Science* **351**:871-875 (2016).

Line 293: this section is very difficult to follow given how the different structures are introduced and compared multiple times, and the intercalating of structure, IC₅₀, mutants (including P510S, its relevance here is not clear). Different complexes, mutants and up and down conformation alternate in a difficult to follow pattern. Twice NSC719801 is mentioned, which is very confusing to the reader. Some comparisons seem less meaningful as 368-380 where two structures in which two variables are changed (mutant AND ligand) are compared.

We apologize for the confusions. P510S, along with N616F, K512N, and F618S, was detected using a specific triazole-resistant HCT116 cell line (**Fig. 3d**). These key residues interacting with the triazole inhibitor are observed in the $p97^{WT}|_{ADP,NSC799462}$ structures (**Fig. 2b**).

Along the line with our study, we used two triazole inhibitors assembled with $p97^{R155H}$: NSC804515 and NSC819701. Possibly due to different attached side chains and fluorine-substituted aromatic rings, $p97^{R155H}|_{NSC804515}$ shows up-NTDs (one structure), but $p97^{R155H}|_{NSC819701}$ has mixed up and down NTDs (two structures).

Compared to the wild-type structure $p97^{WT}|_{NSC799462}$, K615 in $p97^{R155H}|_{NSC804515}$ and $p97^{R155H}|_{NSC819701}$ shows a significant conformational change (**Fig. 5a**). This could be due to the electronic configuration of the substituted 1,2-3-triazole, affecting the packing of P510 (found critical in the cell assay) and the di-fluorobenzene ring.

To understand the change of up and down NTD in $p97^{R155H}$ with triazole inhibitors, we identified the key residues accounted for the conformational changes (**Fig. 5c**). These changes also rearrange the hydrogen bonding network between NTD and D1 domain, thereby regulating up and down NTD conformations (**Fig. 5d**).

We agree with Reviewer 2 and remove the structural comparison of $p97^{R155H}|_{NSC819701}$ with $p97^{WT}|_{NSC799462}$. We update **Fig. 5d** and revise the statement as follows:

(Page 13, Line 373) Remove “... We superimposed the structures of the $p97^{WT}|_{NSC799462}$ and $p97^{R155H}|_{NSC819701}$ (State B) to understand how the R155H mutation impacts the interaction between NTD and D1 domain since both triazole-bound structures show down NTDs (**Fig. 5d**). When bound to NSC819701, the conformations of the K386 and N387 side chains on the helix $\alpha 1$ are changed. ...”

Fig. 5d | Interaction between NTD and D1 domain of the p97^{R155H}_{NSC819701}. Dark green is the NTD of p97^{R155H}_{NSC819701}. Grey surfaces are cryo-EM densities of p97^{R155H}_{NSC819701} in NTD-down conformation. All distances are measured in Å.

Line 387 which findings from NMR spectroscopy?

We apologize for the confusion. We revised the statement to specifically point out the residue N387:

(Page 14, Line 390) From “... *The interaction of R155H with the surrounding residues corroborates the previous findings identified by NMR spectroscopy.*⁴⁹ *In addition, ...*”

To “... *The interaction of the R155H with N387 corroborates the previous findings identified by NMR spectroscopy.*⁴⁹ *In addition, ...*”

Figure 3c: relevant residues should be labeled

We thank for Reviewer 2's suggestion. The revised Figure 3c has the relevant residues labeled.